# Sodium Reduction in Traditional Dry-Cured Pork Belly Using Glasswort Powder (*Salicornia herbacea*) as a Partial NaCl Replacer

**DOI:** 10.3390/foods11233816

**Published:** 2022-11-26

**Authors:** Iasmin Ferreira, Ana Leite, Lia Vasconcelos, Sandra Rodrigues, Javier Mateo, Paulo E. S. Munekata, Alfredo Teixeira

**Affiliations:** 1Centro de Investigação de Montanha (CIMO), Instituto Politécnico de Bragança, Campus de Santa Apolónia, 5300-253 Bragança, Portugal; 2Laboratório Para a Sustentabilidade e Tecnologia em Regiões de Montanha, Instituto Politécnico de Bragança, Campus de Santa Apolónia, 5300-253 Bragança, Portugal; 3Departamento de Higiene y Tecnología de Los Alimentos, Facultad de Veterinaria, Campus Vegazana S/N, 24007 León, Spain; 4Escola Superior Agrária, Instituto Politécnico de Bragança, Campus de Santa Apolónia, 5300-253 Bragança, Portugal; 5Centro Tecnológico de La Carne de Galicia, Rúa Galicia N 4, Parque Tecnológico de Galicia, San Cibrao Das Viñas, 32900 Ourense, Spain

**Keywords:** glasswort, halophytes, sodium reduction, dry-cured pork belly, color, fatty acid profile, sensory profile

## Abstract

Sodium chloride (NaCl) is a key ingredient in the processing of traditional dry-cured meat products by improving microbial safety, sensory attributes and technological properties. However, increasing concern about the consumption of sodium and health has been supporting the development of low-sodium meat products. Several strategies to reduce sodium in dry-cured meat product have been tested, although the followed approaches sometimes result in undesirable characteristics concerning flavor, texture and mouthfeel. The use of halophytic plants such as glasswort (*Salicornia herbacea*) in food matrices has been suggested as a novel strategy to reduce sodium content, due its salty flavor. The main aim of the present study is to produce traditional dry-cured pork bellies from the Bísaro breed using glasswort as a NaCl partial replacer, and compare it with dry-cured bellies salted either with NaCl or a mix of NaCl + KCl. Control bellies (BC) were salted with 100% of NaCl, the second formulation (BK) had 50% of NaCl and 50% of KCl, and the third formulation (BG) had 90% of NaCl and 10% of glasswort powder (GP). After production, the bellies were evaluated for a_w_, pH, CIELab coordinates, weight loss, proximal composition, TBARS, collagen and chloride contents, fatty acid profile and sensory attributes. The use of BG in dry-cured pork bellies did not affect processing indicators such as weight loss, a_w_ and pH. Concerning CIELab, only the coordinates L* and hue angle from the external surface color of BG were statistically different from BC and BK. As expected, ash and NaCl contents differed from BG to the other two formulations. SFA and indexes AI and TI were lower, whereas the MUFA and h/H ratio were higher in BG than other treatments, leading to a product with a healthier lipid profile. The sensory evaluation revealed differences in appearance, taste and flavor among treatments, but did not indicate any negative effects of BG in the product attributes. This study reinforces the potential of BG as a natural sodium reducer for the production of traditional dry-cured pork bellies.

## 1. Introduction

The use of NaCl in the preservation of meat began in ancient times, when salting was used to preserve food for posterior consumption. The advances in knowledge about the use of NaCl revealed its technological, safety, and sensorial roles in dry-cured meat products. More specifically, sodium has an important effect in the control of endogenous enzymes involved in protein and lipid degradation, leading to expected texture and sensory properties during processing. Microbial inhibition is a key effect associated with NaCl by reducing aw activity. In terms of sensory properties, NaCl is inherently associated with the perception of saltiness and has an important effect in the enhancement of food flavor [1,2].

The use of sodium chloride (NaCl) has nowadays become more rational, and is one of the hardest challenges concerning processed food products. The concerns to reduce the content of salt in the food industry started with an awareness that excessive consumption of sodium is a real risk factor for cardiovascular diseases [3,4]. With the course of society’s development, there was a steady increase in the awareness towards food healthiness by consumers and health-related public organizations. Around the 1980s, emerging strategies for the regulation of salt content in processed products started to take place [1,5]. Several meat products such as dry-cured meat contains relatively high sodium concentration. This represents a concerning issue from a health perspective, leading to a poorer reputation and a loss of consumers [6]. In this context, the meat industry is making efforts towards meat products with a low sodium content without affecting their sensory and technological quality. Apart from approaches to remodel consumers’ taste to reduce sodium content in food products, the main strategy for the meat industry is to partially replace the NaCl with non-sodium containing salts such as calcium, potassium and magnesium salts [7,8].

Dry-cured meat products have distinct sensory aspects due to the use of salt and nitrites in their formulations. The rising interest in sodium-reduced meat products, as well as clean label meat products containing natural compounds instead of additives, have brought the possibility of applying new strategies for this challenge, resulting in the presence of innovative and reformulated products in the market [9]. Among these strategies, the use of powdered plant raw materials and extracts (able to provide a salty flavor, along with additives that can offer similar technological traits from NaCl) have been suggested [10,11,12]. The reformulation (including the use of ingredients from algae and plant origin) appears to have a huge potential, although it is necessary to take into consideration their effect in products color, flavor, and others key aspects. In the case of bacon-style dry-cured products, which have a robust reputation among consumers worldwide, recent research has been aiming at reducing salt and nitrites by means of reformulating and modifying the curing methods and treatments [8,13,14].

*Salicornia herbacea* or glasswort is a halophytic plant that grows near seawater and is often mentioned as “green salt” due to its characteristic salty flavor. Consequently, its potential use in the food industry have been studied. The plant can be processed into an extract or powder that is applied in some products such as rice cake, tofu, beef jerky, sausages and more [9,15,16,17]. Moreover, studies also have shown that glasswort powder (GP) contain bioactive compounds, antioxidants, fibers and minerals [11,15].

Considering that the use of natural ingredients can bring favorable flavor, texture, color, moisture and yield into reduced sodium foods [18,19], and the limited studies published on dry-cured meat products produced with BG, the aim of this study is to produce and evaluate the physicochemical and sensory characteristics of a sodium-reduced traditional dry-cured pork belly prepared using glasswort (without nitrates/nitrites) in the curing mix.

## 2. Materials and Methods

### 2.1. Formulation and Production of Dry-Cured Pork Bellies

Fresh pork bellies were obtained from Bísaro pigs weighting approximately 135 ± 5 kg, reared in a mixed production system, and slaughtered at the municipal slaughterhouse of Bragança in Portugal. Three mixtures of salts were prepared, one with NaCl (control; BC), another with both NaCl and KCl (50% each; BK), and the other with both NaCl and GP (90% and 10%, respectively; BG). The powdered glasswort was purchased from Salivitae Lda. (Algarve, Portugal) with the following nutritional information: Proteins 13.9 g/100 g; Fatty Acids (FA) 0.53 g/100 g; Na 18.9 mg/100 g; Cl 14.5 mg/100 g; and K 1.39 mg/100 g. The bellies were defrosted at 4 °C for 48 h and then dry-cured using a traditional method with 4% salt concentration (*w*/*w*) for BC and BK, and 2% for BG.

The production of bellies was carried in the dependencies of Agriculture School of Polytechnic Institute of Bragança (Carcass and Meat Quality Laboratory, Portugal). Figure 1 illustrate the processing of dry-cured pork bellies. Briefly, the defrosted bellies were deboned and cut to have a homogenous right-angle shape and weight of approximately 1 kg. Then, the pork bellies were placed in trays and manually rubbed with the salt mixtures. After 48 h, the excess of salt was removed from the bellies’ surface with warm (40 ± 5 °C) tap water and refrigerated for 60 h. Two heating steps were then applied with 48 h of resting period between them, consisting of heating the bellies for 1 h at 90 °C in an oven with relative humidity (RH) of 70–80% (BriCANTEL, Bragança, Portugal). Afterwards, the bellies were air dried for 1 week (10 ± 5 °C/30–35% RH). Figure 2 shows the bellies at the end of the process. Finally, the bellies were vacuum packaged at 4 °C for 7 days.

After storage, the bellies were unpacked and approximately a 50 g sample from each dry-cured belly was taken, the skin was removed, and the sample was homogenized with a BUCHI Mixer B-400 before analysis. Two replications were manufactured at different times and for each replicated batch, three samples of each treatment were randomly selected in a total of 18 bellies, and each sample was analyzed in triplicate for each physicochemical analysis.

### 2.2. Dry-Cured Pork Bellies Technological Quality Traits and Composition Analysis

Weight loss (WL) was evaluated after both the salting stage (T1) and at the end of the process (T2) by weighting fresh, salted and dry bellies indicated as percentage (Equation (1)).
(1)$$\mathrm{WL}\;\left(\%\right)=\frac{\mathrm{fresh}\;\mathrm{weight}-\mathrm{dry}\;\mathrm{weight}}{\mathrm{fresh}\;\mathrm{weight}}$$

Product pH was measured with a Crison 507 pH-meter (Crison Instruments, Barcelona, Spain). The penetration probe electrode (52–32 puncture) was inserted in the lean sections of the bellies after thawing and at the end of the dry curing process. The assay was carried out following the standard method NP-ISO 3441/2008 [20]. Water activity (a_w_) was determined with approximately 10 g piece of cut belly (after thawing and at the end of process) using a probe HigroPalmAw1 Rotronic 8303 (Bassersdorf, Switzerland), according to AOAC [21].

Instrumental color was evaluated using Lovibond RT Series—SP62 spectrophotometer (The Tintometer Limited, Wiltshire, England). The data was measured at 10 nm intervals with reflectance in the range 400–700 nm in the CIELab coordinates [22]. Each belly was evaluated both on the upper surface and on the transversal cut (measured 15 min after cutting) for lightness (L*), redness (a*) and yellowness (b*). Chroma (saturation index—C*) and hue angle were also calculated, according to the Equations (2) and (3), respectively.
(2)$$C*=\sqrt{\left({a*}^{2}+{\;b*}^{2}\right)}$$
(3)$$\mathrm{Hue}\;\mathrm{angle}={\;\tan}^{-1}\left(\frac{a*}{b*}\right)$$

The oxidation status of the bellies, assessed by the thiobarbituric acid reactive substances (TBARS), was performed by extracting malondialdehyde (MDA) as stated in the standard method NP-ISO-3356/2009 with modifications. Briefly, the extraction procedure was carried out by homogenizing the sample (≈2 g) in distilled water (20 mL). Then, a 25% aqueous solution of TCA was added, and the mix was centrifuged for 15 min at 12,000 rpm. Lipid oxidation status were indicated as mg of MDA/kg of sample [23].

The standard method NP-ISO-1614/2002 was used to determine the moisture content [24]. Briefly, the sample (≈3 g) was homogenized with ethanol (5 mL). The solvent was completely evaporated by keeping the mixture at 70 °C. Afterwards, the mix was oven-dried until constant weight at 103 ± 2 °C. The content of ash was determined using the same samples from moisture determination by incinerating at 550 °C, according to the protocol NP-ISO-1615/2002 [25]. Protein content was assessed using the Kjeldahl method in accordance with the method NP-ISO-1612/2006 [26]. Myoglobin was quantified by spectroscopy (Spectronic Unicam 20 Genesys) at 512 nm following the protocol described by Hornsey [27]. Results were indicated as mg myoglobin/g fresh muscle. The determination of hydroxyproline was used to indicated the collagen content by following the standard protocol NP 1987/2002 protocol [28]. The chloride content, expressed as a mass percentage of sodium chloride, was obtained following the Portuguese Standard NP 1845/1982 [29]. Total lipid content was determined using 25 g of the homogenized sample and quantified following the method described by Folch et al. [30].

The fatty acid (FA) profile was determined using ≈50 mg of the fat. Once obtained, FAs were transesterified according to Shehata et al. [31] with the adjustments described by Domínguez et al. [32] and details pointed out by Teixeira [33] to indicate the content of fatty acids as g of FA/100 g of total FAs. The nutritional quality of FAs was determined by the n-6/n-3 and polyunsaturated/saturated FA (PUFA/SFA) ratios [34]. The thrombogenicity (TI) and atherogenicity (AI) indexes were calculated as indicated in Equations (4) and (5) and described by Ulbricht and Southgate [35]. The hypocholesterolemic and hypercholesterolemic fatty acids (h/H) ratio was also calculated, as indicated in Equation (6) according to Santos-Silva, Bessa & Santos-Silva [36].
(4)$$\mathrm{AI}=\frac{C12:0+4\;\times \;C14:0+C16:0}{\sum^{\;}\mathrm{MUFA}+\sum^{\;}\mathrm{PUFA}}$$
(5)$$\mathrm{TI}=\frac{C\;14:0+C16:0+C18:0}{0.5\times \sum^{\;}\mathrm{MUFA}+0.5\times \sum^{\;}\mathrm{PUFA}\;n-6+3\times \sum^{\;}\mathrm{PUFA}\;n-3+\frac{\mathrm{PUFA}\;n-3}{\mathrm{PUFA}\;n-6}}$$
(6)$$\frac{h}{H}=\frac{C18:1n-9+\;C18:2n-6+C20:4n-6+C18:3n-3+\;C20;5-n3+C22:5n-3+C22:6n-3}{C14:0+C16:0}$$

### 2.3. Sensory Analysis of Dry-Cured Pork Bellies

A sensory analysis of the dry-cured pork bellies was performed by a trained taste panel in three different sessions. The attributes evaluated in the dry-cured bellies were related to appearance, meat color, fat color, and meat/fat ratio; texture (crunchiness, juiciness and toughness); basic tastes (salty, sweet, sour and bitter); and aroma and flavor descriptors and intensity. For the evaluation of the raw product, the slices were placed in white plastic plates and the evaluated attributes were those related to appearance and aroma. For the cooked product, the slices were previously placed in aluminum foil and cooked in an oven at 170 ± 5 °C for 14 min (turning the slices upside down at half time), and all of the above-mentioned attributes were evaluated. The panel consisted of nine assessors. Their selection followed the Portuguese Standard protocol [37] for recruitment, selection, training stages for evaluation of meat and meat products. The specific training of panelists was carried out at the Polytechnic Institute of Bragança (Sensory Analysis Laboratory). The temperature of the training room was maintained at 20 ± 2 °C, and the testing followed the standard guidelines, with RH at 50 ± 5% [38]. Room luminosity was homogeneous for all booths with white lights on. Water and apple slices were provided to cleanse the palate between samples. The samples were sliced with 2 mm thickness, using a vertical slicer (FAC s.r.l., Cavaria, Italy), and each slice (9 per session) was randomly coded and individually given to the panelists.

### 2.4. Statistical Analysis

The differences among dry-cured pork belly treatments were evaluated using the JMP^®^ Pro 16.0.0 statistical packaged (SAS Institute Inc.©, Cary, NC, USA) by fitting a standard least square model. Significant (*p* < 0.05) differences indicated by ANOVA were ranked for Tukey´s HSD test with significance levels of *p* < 0.01 or *p* < 0.001.

The sensory analysis data was statistically evaluated with XLStat program (Addinsoft, New York, NY, USA). Firstly, the characterization of the product was performed, and after, in order to minimize differences and identify agreement between panelists, a generalized Procrustes analysis (GPA) was performed. The consensus among the nine assessors matched the data matrices of three (bellies formulations) by seventeen (sensory attributes) factors. The results were displayed in graphs.

## 3. Results and Discussion

### 3.1. Technological Traits and Compositional Analysis of Dry-Cured Bellies

Figure 3 shows the results for WL during T1 (after salting) and T2 (end of process). The formulations present no significant differences, meaning that the modifications of NaCl content in BG and BK did not affect the losses during the processes of salting and air drying. Throughout the salting and drying stages, there was a considerable weight reduction of the bellies that was larger in the salting (due to the loss of fluids resulting from the hygroscopic characteristic of NaCl) than in the drying (due to evaporation) stage. This result was seen in previous studies with dry-cured bacon [2].

The values obtained for pH of the dry-cured pork bellies are shown in Figure 4. The pH values of the fresh bellies differed among treatments, being a few hundredths higher in the BK group than in other groups. The pH values decreased through the process by approximately one tenth, and the effect of treatment on the final pH became insignificant (*p* = 0.056). In contrast to our results, Jin et al. [2] found neither changes nor small increments in pH during the dry-curing process of the bellies. However, Song et al. [39] found the pH of dry-curing bellies to decrease during storage, and attributed this change to the growth of lactic acid bacteria. The growth of these bacteria could be the reason for the decrease found in the present study.

Results for a_w_ are presented in Figure 5. It is possible to observe the differences among bellies from each treatment, before the beginning of the process can be attributed to random effects based on intrinsic variations between animals. However, no statistical differences were found in dry-cured bellies after processing. Therefore, the final a_w_ suggests that using the mixture BG (2%) had a performance comparable to NaCl (4%) and mix of NaCl + KCl (4%) as an aw-reducing agent. 

Seong et al. [14] also found that the use of GP did not affect the decrease of aw in dry-cured ham. However, the a_w_ of the dry-cured belly in the present study is lower than that reported by Kartaliovic et al. [40] for a traditional Balkan dry-cured belly. This difference is probably explained by the more intense drying in the Portuguese dry-cured belly.

Differences in the a_w_ of the dry-cured bellies depends on the lean percentage, salts concentration, and drying conditions [41]. Moreover, the final a_w_ values of the dry-cured bellies from this study (c.a. 0.8) are into the range of the intermediate moisture meat products and confers the product microbial stability [41].

Table 1 shows results for CIELab coordinates in dry-cured pork bellies. Statistical differences were only found in the surface measurements, which can affect the consumer’s buying intention, whereas there was no impact of BG use for the color of the transversal cut. Surface lightness was affected by the BG, since it showed a significantly lower value than BC and BK. The same effect among treatments was observed for hue values. In another experiment with beef jerky, the use of BG resulted in lower lightness, redness and yellowness, which was attributed to the presence of chlorophyl and other pigments in GP [15,42]. Similarly, Jeong et al. [43] also observed differences in color properties (particularly L* and a* values) of pork loin ham salted with GP. Additionally, the use of GP as a salt replacer in other salted food products also affected CIELab coordinates [44].

Table 2 shows the composition of dry-cured pork bellies. Moisture content was not affected by the treatment, which agrees with what has been observed in other studies using GP in the production of intermediate-moisture meat products [15,43]. Ash content differed statistically, being lower in BG than other treatments. This finding suggests that less amounts of salts are diffused in the dry-cured belly, since a direct relation between NaCl and ash contents in dry-cured meat is expected [15,45]. Collagen content differed between formulations BC and BG. Although the formulations used in this experiment would not cause this difference, this variation among treatment can be due to different animal characteristics. Regarding chloride content, differences were found between BC and BG.

TBARS values did not differ across different treatments and were in the range of those found by Jin et al. [46] in dry-cured bacon (i.e., from 0.2 to 0.4). In addition, Jo et al. [44] obtained a similar relation for TBARS values in a different type of product with GP. The potential antioxidant effect of GP mentioned by previous authors [47,48] could not be observed either in the dry-cured pork bellies from this study, or in those from that study [44]. This effect might perhaps be seen after longer storage periods.

The FA profile of dry-cured pork bellies is shown in Table 3. In BG bellies it is possible to observe significant lower values for some SFAs (C15:0, C16:0, C17:0 and C18:0) in relation to BC and BK. BG treatment was also associated with significant high values of oleic acid (C18:1n-9) and heneicosanoic acid (C21:0) in comparison to BC and BK. Differences were also be observed in C20:1n-9 content, although for this FA, BC bellies differed from BK and BG. 

Lipidic quality indicators were also significantly affected by treatments. In the case of SFA, BG bellies presented the lowest values among all treatments. Conversely to what was observed in our experiment, Seong [15] did not find differences in ham produced with GP for this characteristic. This difference may be due to the different anatomic region used or the amount of NaCl replaced. For MUFA, BG had the highest values among all formulations, indicating that BG was able to preserve these fatty acids during processing. Consequently, the same effect could be observed for the UFA/SFA ratio. Regarding the indexes, there were significant differences among treatments: bellies cured with BG had the lowest AI and TI values and the highest h/H ratio than those cured with NaCl (BC) or NaCl and KCl (BK) mixture.

### 3.2. Sensory Analysis of Dry-Cured Bellies

From the attributes used to describe differences among dry-cured bellies, five were assessed in the raw product and twelve in the cooked product. Table 4 shows the attributes tested and the *p*-values resulting from the sensory analysis.

A GPA was carried out from the panel’s responses. Despite the panelist training process, some variability in the performance among the panelists was observed. In Figure 6a,b is possible to observe the GPA residuals values for formulation and panelists, respectively. Concerning the formulations, the BG bellies showed the lowest residuals (1.450), therefore a higher level of consensus among panelists was obtained for the sensory analysis. Regarding the panelists, those identified by the numbers 2, 3, 6, 7, 8 and 9 had a good agreement for the attributes, while, 1, 4 and 5 showed higher variation. 

Table 5 shows the scaling factors and percentage of variation of the two principal factors (F1 and F2) for each panelist. The panelists 2, 4, 6, 8 and 9 used a wider range of the scale, as shown by a scaling factor value above 1. 

The biplot obtained from the GPA is presented in Figure 7. For the product sensory characterization, 100% of the variability was explained by the factors F1 (78.16%) and F2 (21.84%). Conversely, other studies exploring the correlations among sensory attributes with two dimensions reported total variance values of 63.21% for dried pork meat produced with different levels of salt [49], and 76.11% in dry-cured loins cured with different salt replacers [50].

The attributes that best correlated with factor 1 (F1) in the raw samples were meat/fat ratio (0.986), fat color (−0.779) and aroma intensity (−0.835), whereas cooked samples had a high correlation with chewiness (0.981), sweetness (0.962), meat/fat ratio (0.940), crunchiness (−0.767), sour (−0.894), saltiness (−0.939), juiciness (−0.981) and flavor persistency (−0.998). Particularly for factor 2 (F2) axis, the raw samples showed a high correlation with meat color (0.966) and cooked samples were mainly correlated with bitterness (0.951), flavor intensity (0.894), meat color (0.793), and fat color (−0.955). Regarding the formulations, BC and BK were placed in the opposite direction to BG on the F1 axis. Moreover, BC was located on the negative side of F2, BK appeared in the positive quarter, and BG (although close to 0) was also placed in the negative quarter of F2. More specifically, the attribute that was closest to BC and most correlated to this group was crunchiness, for BK it was the flavor intensity, and for BG it was raw aroma intensity (raw).

In agreement with other studies [51,52], and as expected, the saltiness was more perceived in and correlated with the BC bellies. Interestingly, saltiness also correlated with sour taste. However, to the best of our knowledge, no previous studies have related these two sensory attributes in reduced sodium dry-cured meat. Juiciness was related to BC and thus with saltiness (NaCl content), which agrees with previous studies of pork meat, where a relation between NaCl content and the juiciness of the dry-cured meat was also reported [51,53]. In BK bellies (as result of KCl presence) a stronger bitterness was expected, as seen in other studies [52]; however, this attribute was placed in the opposite quarter. Finally, for the BG bellies, the two attributes most closely related were aroma intensity in the raw samples and chewiness. For the first one (due to GP), it was expected that the panelists would perceive an earthy or plant aroma; although in the aroma identification there was no mention of it for either the raw or cooked samples. For chewiness, several studies indicate textural changes in the meat as a result of salt reduction, but previous research on the use GP in meat products meat has not related its use to texture alterations [15,42].

## 4. Conclusions

Using GP to partially replace NaCl in salt reduced dry-cured pork bellies did not affect the drying process, indicated by the results of WL and a_w_; the pH values also stayed in the acceptable range. There were small instrumental color variations in the dry-cured belly surface, since the luminosity of the product diminished. Nevertheless, the interior color of the product was not affected by any of the sodium replacers. Regarding the composition, the parameter that was affected specifically in BG was the ash content, which was lower than the Cl content and in BG than all formulations used. The lipidic profile was positively affected by the GP (comprising less SFA and more MUFA), leading to a product with slightly healthier lipid indexes (AI, TI, and h/H ratio). The sensory evaluation by panelists was able to characterize the product explaining 100% of variability with two dimensions, and identified the main sensory attributes of each formulation group.

In conclusion, the use of GP seems a feasible approach for partial NaCl replacement. It can be applied in traditional dry-cured bellies without a major impact in the physical chemical and sensory characteristics. Notwithstanding, further studies should be performed in order to explore the full aptitude of glasswort in meat products as a naturally sourced sodium replacer.

## Figures and Tables

**Figure 1 foods-11-03816-f001:**
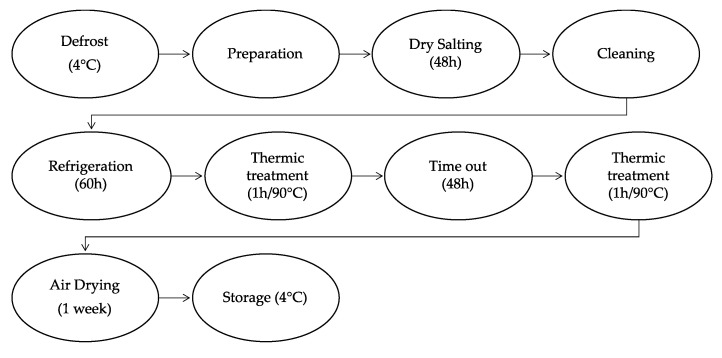
Process chart of dry-cured pork bellies.

**Figure 2 foods-11-03816-f002:**
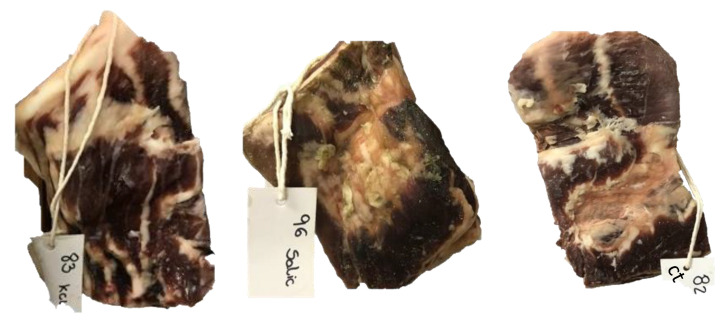
Dry-cured bellies according to treatments (BK: 83; BG: 96; BC: 82).

**Figure 3 foods-11-03816-f003:**
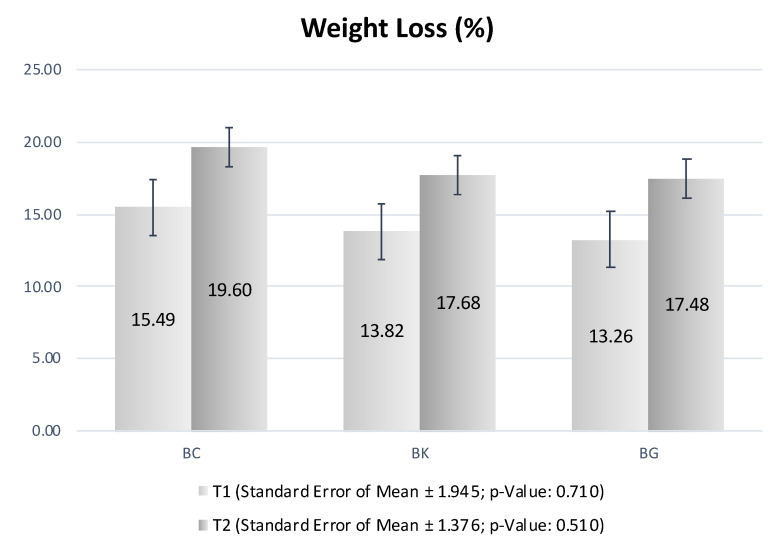
Weight loss (%) of pork bellies after salting (T1) and at the end of process (T2).

**Figure 4 foods-11-03816-f004:**
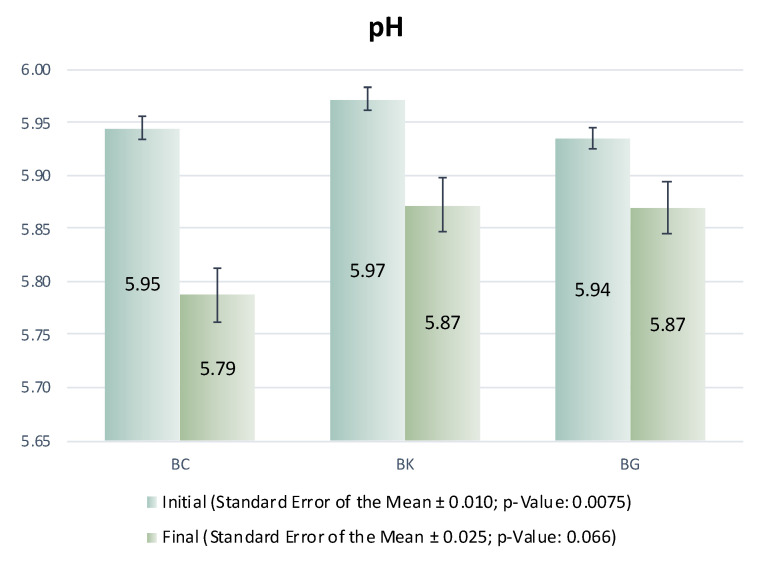
pH of pork bellies after thawing (Initial) and at the end of process (Final).

**Figure 5 foods-11-03816-f005:**
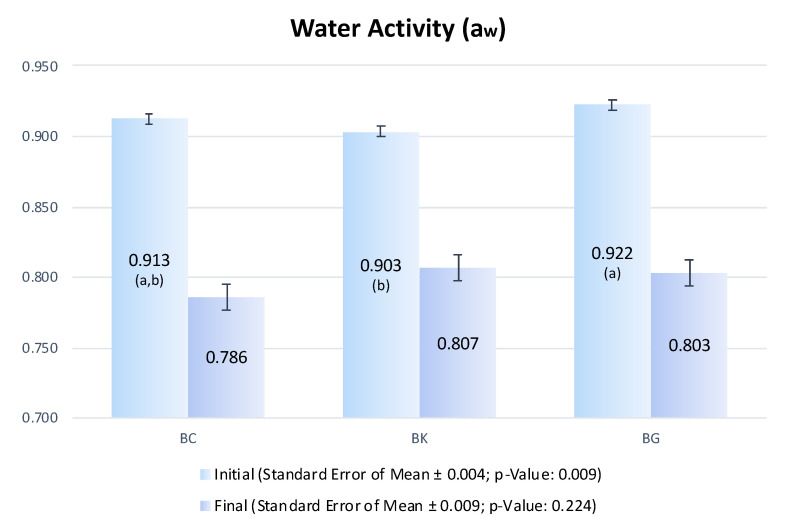
Water activity (a_w_) of pork bellies after unfreezing (Initial) and in dry-cured pork bellies (Final). Different letters (a–b) inside brackets shows statistical differences between pork bellies.

**Figure 6 foods-11-03816-f006:**
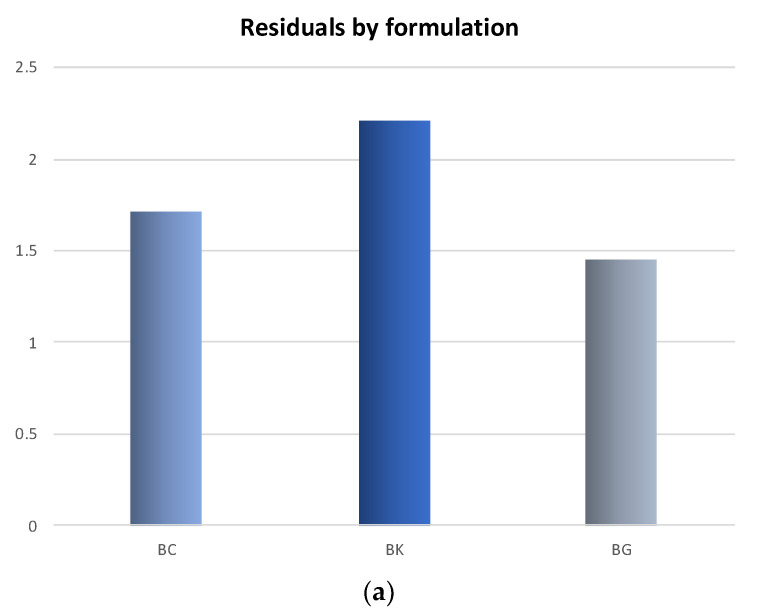
(**a**) GPA residual values by formulation; (**b**) GPA residual values by panelist.

**Figure 7 foods-11-03816-f007:**
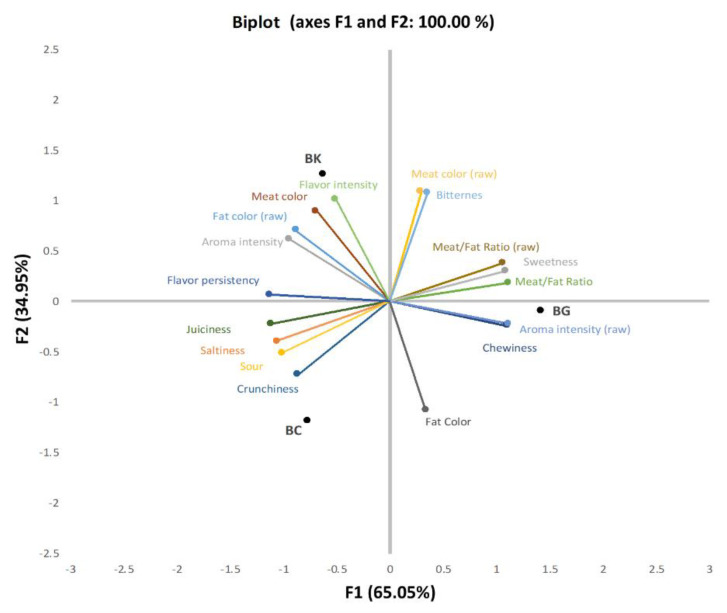
Biplot correlating sensory attributes in the plot factor 1 (F1) and 2 (F2) and dry-cured belly coordinates. Attributes determined in raw samples are marked with raw between brackets.

**Table 1 foods-11-03816-t001:** CIELab coordinates of dry-cured pork bellies with sodium reduction.

Location	Parameter	BC	BK	BG	SEM	*p*-Value
Surface	L*	50.45 ^a^	49.12 ^a^	37.99 ^b^	3.175	0.017
	a*	5.99	6.49	6.61	0.870	0.870
	b*	12.17	11.37	9.09	1.486	0.327
	Hue angle	64.51 ^a^	57.73 ^a,b^	50.47 ^b^	3.840	0.048
	Chroma	13.63	13.30	11.74	1.548	0.657
Transversal cut	L*	68.55	70.28	72.07	1.626	0.323
	a*	7.65	7.40	8.46	0.713	0.555
	b*	9.20	8.56	8.94	0.483	0.645
	Hue angle	50.87	50.49	47.27	1.777	0.302
	Chroma	12.03	11.39	12.35	0.788	0.686

SEM: Standard Error of the Mean; ^a,b^ mean values in the same row not followed by a common subscripted letters differ significantly (*p* < 0.05; Tukey test).

**Table 2 foods-11-03816-t002:** Physicochemical parameters of dry-cured pork bellies with sodium reduction.

Parameter	BC	BK	BG	SEM	*p*-Value
Moisture (%)	30.31	30.10	31.86	1.359	0.582
Ashes (%)	4.86 ^a^	4.34 ^a^	3.03 ^b^	0.200	<0.0001
Protein (%)	14.26	13.44	13.99	1.064	0.859
Fat (%)	48.18	56.73	55.31	3.974	0.286
Collagen (%)	1.87 ^a^	1.48 ^a,b^	1.14 ^b^	0.151	0.009
NaCl (%)	4.32 ^a^	3.40 ^a,b^	2.62 ^b^	0.329	0.006
TBARS (mg of MDA/kg of sample)	0.29	0.24	0.31	0.023	0.178

SEM: Standard Error of the Mean; ^a,b^ mean values in the same row not followed by a common subscripted letters differ significantly (*p* < 0.05; Tukey test).

**Table 3 foods-11-03816-t003:** Lipid profile of dry-cured pork bellies with sodium reduction.

Fatty Acids (%)	BC	BK	BG	SEM	*p*-Value
Luric acid (C12:0)	0.028	0.019	0.027	0.007	0.599
Myristic acid (C14:0)	1.122 ^a^	1.068 ^b^	1.050 ^b^	0.024	0.018
Pentadecanoic acid (C15:0)	0.034 ^a^	0.027 ^a^	0.011 ^b^	0.006	0.021
Palmitic acid (C16:0)	25.165 ^a^	24.743 ^a^	23.547 ^b^	0.323	0.006
Palmitoleic acid (C16:1n-7)	2.504 ^a^	2.303 ^a,b^	2.117 b	0.094	0.029
Heptadecanoic acid (C17:0)	0.267 ^a^	0.251 ^a^	0.206 ^b^	0.013	0.012
cis-10-Heptadecenoic (C17:1n-7)	0.288 ^a^	0.275 ^a,b^	0.225 ^b^	0.018	0.047
Stearic acid (C18:0)	11.634 ^a^	11.472 ^a^	10.971 ^b^	0.182	0.045
Elaidic acid (9t-C18:1)	0.227	0.239	0.252	0.013	0.418
Oleic acid (C18:1n-9)	50.406 ^b^	50.778 ^b^	52.511 ^a^	0.524	0.022
Linolelaidic acid (9t,12t-C18:2)	0.005	0.000	0.003	0.002	0.412
Linoleic acid (C18:2n-6)	7.039	7.228	7.248	0.267	0.831
Arachidic acid (C20:0)	0.196	0.206	0.205	0.006	0.461
ɣ-Linolenic acid (C18:3n-6)	0.016	0.009	0.013	0.006	0.717
cis-11-Eicosenoic acid (C20:1n-9)	0.134 ^b^	0.433 ^a^	0.590 ^a^	0.136	0.046
Linolenic acid (C18:3n-3)	0.268	0.276	0.275	0.012	0.874
Heneicosanoic acid (C21:0)	0.006 ^b^	0.004 ^b^	0.029 ^a^	0.004	0.001
cis-11,14-Eicosadienoic acid (C20:2n-6)	0.297	0.318	0.333	0.020	0.474
Behenic acid (C22:0)	0.039	0.034	0.044	0.007	0.531
cis-8, 11, 14-Eicosatrienoic acid (C20:3n-6)	0.042	0.038	0.042	0.007	0.917
cis-11,14,17-Eicosatienoic acid (C20:3n-3)	0.036	0.034	0.044	0.007	0.555
Tricosanoic acid (C23:0)	0.127	0.140	0.192	0.025	0.176
Arachidonic acid (C20:4n-6)	0.042	0.045	0.000	0.024	0.349
Nervonic acid (C24:1n-9)	0.064	0.058	0.061	0.011	0.912
cis-4,7,10,13,16,19-Docosahexaenoic acid (C22:6n-3)	0.015	0.003	0.012	0.005	0.250
Σ SFA	38.619 ^a^	37.963 ^a^	36.278 ^b^	0.433	0.003
Σ MUFA	53.622 ^b^	54.086 ^b^	55.754 ^a^	0.526	0.023
Σ PUFA	7.758	7.952	7.968	0.301	0.861
Σ UFA/Σ SFA	1.593 ^b^	1.636 ^b^	1.758 ^a^	0.031	0.003
Σ n-6	7.436	7.639	7.635	0.288	0.851
Σ n-3	0.318	0.313	0.331	0.019	0.772
Σ n-6/Σ n-3	23.857	25.151	23.225	1.392	0.615
AI	0.484 ^a^	0.468 ^a^	0.436 ^b^	0.009	0.005
TI	1.205 ^a^	1.172 ^a^	1.088 ^b^	0.021	0.003
h/H	2.203 ^b^	2.268 ^b^	2.444 ^a^	0.048	0.005

Σ SFA: sum of saturated fatty acids; Σ MUFA: sum of monounsaturated fatty acids; Σ PUFA: sum of polyunsaturated fatty acids; Σ UFA: sum of unsaturated fatty acids; AI: atherogenicity index; TI: thrombogenicity index; SEM: Standard Error of the Mean; ^a,b^ mean values in the same row not followed by common subscripted letters differ significantly (*p* < 0.05; Tukey test).

**Table 4 foods-11-03816-t004:** Attributes used for sensory evaluation of dry-cured pork bellies: raw (marked as raw in brackets) and cooked (rest of attributes), and *p*-values obtained from the comparison of treatments.

Attributes	Scores	*p*-Values
Appearance		
Meat/Fat ratio (raw)	3.766	<0.0001
Meat/Fat ratio	3.742	<0.0001
Fat Color (raw)	0.960	0.168
Fat Color	1.810	0.035
Meat color (raw)	0.557	0.289
Meat color	0.317	0.376
Taste		
Sweetness	0.677	0.249
Saltiness	1.088	0.138
Bitterness	−0.112	0.545
Sour	−0.341	0.634
Texture		
Juiciness	0.615	0.269
Chewiness	0.367	0.357
Crunchiness	0.273	0.392
Odour and flavor		
Flavor intensity	1.831	0.034
Aroma intensity (raw)	1.640	0.051
Flavor persistency	−0.150	0.559
Aroma intensity	−1.659	0.951

**Table 5 foods-11-03816-t005:** Scaling factors and percentage of variation explained by the first two principal components (F1 and F2) for each panelist for the dry-cured pork belly sensory analysis.

Panelist	Scalling Factor	F1	F2
1	0.869	39.492	60.508
2	1.313	66.779	33.221
3	0.968	72.708	27.292
4	1.236	52.112	47.888
5	0.671	85.026	14.974
6	1.138	65.654	34.346
7	0.895	59.712	40.288
8	1.427	74.844	25.156
9	1.161	64.125	35.875

## Data Availability

Data is contained within the article.

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
