# Peer review of "Sodium Reduction in Traditional Dry-Cured Pork Belly Using Glasswort Powder (Salicornia herbacea) as a Partial NaCl Replacer"

_foods, 2022, doi:10.3390/foods11233816_

Round 1
Reviewer 1 Report
The submitted article for review corresponds to the subject of the authoritative scientific journal Foods (ISSN 2304-8158)
Manuscript ID foods-2042811
Type Article
Title Sodium reduction in traditional dry-cured pork belly using glasswort powder (Salicornia herbacea) as common salt replacer
Dear authors:
A good scientific work of the publication is presented.
The title is good and quite specific and attracts the reader's interest so that it is easy to understand.
The topic of scientific research is quite interesting and relevant.
The scientific article is logically built, corresponds to the principles of presenting scientific information and research.
I think that the article used enough tables and illustrations.
There are some minor suggestions:
Abstract:
I recommend the authors to expand, increase the number of words. This is necessary to increase the attention of readers (researchers and practitioners) to the research conducted by the authors.
keywords
I suggest adding a few terms to expand the search
Introduction
The information in this section is too short and narrow. The problem raised by the authors is of scientific interest.
I recommend significantly improving (adding, expanding).
Materials and Methods
Sufficiently detailed, in accordance with the purpose of research.
Structured quite well, at all stages in accordance with the research plan of the authors.
Results
Set out in sufficient detail, structured, debatable.
Presentation at a fairly good scientific level.
Correspond to the objectives of scientific research.
However, I note the negligence in the design of the illustrations presented in the manuscript. They need to be improved.
It is possible to present the results in figures 4 and 5 in a slightly different way.
Use different colors in the chart (Figure 6).
The lines should be clearly marked in figure 7.
Discussion
interesting, convincing, clear presentation
Conclusions
It is necessary to significantly supplement and expand in accordance with the purpose and objectives of your research.
The authors summed up the results of the enormous scientific work done too briefly.
References
It is necessary for the author to carefully check the design of the list and standardize it to the requirements of a scientific journal.
I note that the authors of the article approached the analysis of the problem quite carefully and used the necessary amount of regulatory documentation (international and national). Some sources are in Portuguese, but the authors did not put this information at the end of the source, for example (In Portuguese). The text is provided in Portuguese and English. Although the text of the article is presented in English.
About 50 percent of information sources (presented in the List of References (bibliographic list) have been published over the past 5 years.

Author Response
Dear reviewer,
We appreciate the time spent with the work presented, all modifications were made following the reviewer's suggestions and comments, and responses to the comments are also attached. Thanks to your recommendations, significant modifications were made throughout the manuscript.
Thank you for your attention.
Abstract
I recommend the authors to expand, increase the number of words. This is necessary to increase the attention of readers (researchers and practitioners) to the research conducted by the authors.
More information was added to the abstract.
keywords
I suggest adding a few terms to expand the search
Terms were modified and added to describe the work performed.
Introduction
The information in this section is too short and narrow. The problem raised by the authors is of scientific interest.
I recommend significantly improving (adding, expanding).
The authors added more information in the introduction.
Materials and Methods
Sufficiently detailed, in accordance with the purpose of research.
Structured quite well, at all stages in accordance with the research plan of the authors.
Results
Set out in sufficient detail, structured, debatable.
Presentation at a fairly good scientific level.
Correspond to the objectives of scientific research.
However, I note the negligence in the design of the illustrations presented in the manuscript. They need to be improved.
It is possible to present the results in figures 4 and 5 in a slightly different way.
Use different colors in the chart (Figure 6).
The lines should be clearly marked in figure 7.
The main goal of the illustrations design was to be simple and comprehensible, sorry for this misunderstanding. Considering the suggestions, the design was modified and the colors too.
Discussion
interesting, convincing, clear presentation
Conclusions
It is necessary to significantly supplement and expand in accordance with the purpose and objectives of your research.
The authors summed up the results of the enormous scientific work done too briefly.
There was a slightly increase of information in conclusion, in order to match with the amount of results obtained.
References
It is necessary for the author to carefully check the design of the list and standardize it to the requirements of a scientific journal.
I note that the authors of the article approached the analysis of the problem quite carefully and used the necessary amount of regulatory documentation (international and national). Some sources are in Portuguese, but the authors did not put this information at the end of the source, for example (In Portuguese). The text is provided in Portuguese and English. Although the text of the article is presented in English.
The references were checked and the information In Portuguese was placed whenever necessary.

Reviewer 2 Report
Manuscript: Sodium reduction in traditional dry-cured pork belly using glasswort powder (Salicornia herbacea) as common salt replacer
This manuscript was interesting for topic. Sodium reduction has a continuous issue worldwide and is being resolved in various ways. But it has some problems in this manuscript. First of all, the title and the contents of this study do not match.
For some reason, the author did not simply set the treatment to the additive ratio of glasswort, but only substituted 10%. So, is this a salt substitute? It seems that accurate opinions about that or corrections are needed. Rather than saying salt substitute, it would be better to modify the title to focus on salt replacement or salt reduction.
In addition, the full name should be written again by distinguishing the title, abstract, and another manuscript.
Abstract: dry salted and dry cured can be synonymous, so authors need to unify this throughout the manuscript.
It is judged that the BG that replaced 10% cannot represent the GP, so the problem of confusing and using BG with GP should be resolved (like as line 27).
It is also necessary to accurately describe the subject and object in the description of the results (eg, the control group has a higher pH than the treatment group).
The conclusions in the abstract are vaguely expressed, and it is important to clarify their intent.
Introduction: The effect of salt on meat products needs further explanation. Paragraphs 1 and 2 both describe the current problems and current status of salt. In order to confirm that there are no problems with sensory and physicochemical properties even if this study is salt replaced, it is judged that it is necessary to explain the state of salt's effect on meat.
Also what does common salt mean?
Materials and methods: For normal dry cured ham, heating conditions do not apply. Therefore, the author needs a prior study on how this was manufactured. You also need to accurately describe this for how many batches and repetitions.
Results and discussion: Line 197-199: It is important to explain the cause of this. We already know about this fact or expectation, but I wonder how to explain this principle in this manuscript. The author needs to look at the considerations in the manuscript as a whole. This is very important. I have a hard time seeing most of the reviews as supporting content. It seems that only the analysis of the prediction was made, not the exact principle analysis. Also, when describing conflicting considerations, describe how they differ.
Other minor issues include:
Fig. 3. In BG, you can infer from the picture that 13.26 above means T1 and 17.48 means T2, but this picture has a problem. Are there any precedents for this being used? Expressing significant differences is insufficient (within figures). Overall, the question arises as to why the figure was produced in this form and not as a simple bar graph.
Expressions such as (BG at 2%) expressed on line 219 need to be rewritten.
Line 224-226: Is the comparison made because it was conducted under the same conditions as this study?
Line 235: Do consumers often buy dry-cured ham as whole meat?
Author Response
Dear reviewer,
All modifications were made according to the reviewer's suggestions and comments, and responses to the comments are also attached. Thanks to your recommendations, significant modifications were made throughout the manuscript.
Thank you for your attention.
This manuscript was interesting for topic. Sodium reduction has a continuous issue worldwide and is being resolved in various ways. But it has some problems in this manuscript. First of all, the title and the contents of this study do not match.
For some reason, the author did not simply set the treatment to the additive ratio of glasswort, but only substituted 10%. So, is this a salt substitute? It seems that accurate opinions about that or corrections are needed. Rather than saying salt substitute, it would be better to modify the title to focus on salt replacement or salt reduction. In addition, the full name should be written again by distinguishing the title, abstract, and another manuscript.
Thanks for your suggestion. The title was modified accordingly, and not the term substitution but that of replacement has been used throughout the text.
Abstract: dry salted and dry cured can be synonymous, so authors need to unify this throughout the manuscript.
The terms were unified and ‘dry cured’ was chosen for this manuscript.
It is judged that the BG that replaced 10% cannot represent the GP, so the problem of confusing and using BG with GP should be resolved (like as line 27).
This phrase was replaced for:
“The partial replacement of salt by GP”
It is also necessary to accurately describe the subject and object in the description of the results (eg, the control group has a higher pH than the treatment group).
The products were correctly addressed in the revised manuscript.
The conclusions in the abstract are vaguely expressed, and it is important to clarify their intent.
More information was added in the abstract.
Introduction: The effect of salt on meat products needs further explanation. Paragraphs 1 and 2 both describe the current problems and current status of salt. In order to confirm that there are no problems with sensory and physicochemical properties even if this study is salt replaced, it is judged that it is necessary to explain the state of salt's effect on meat.
More information was mentioned about the use of salt in meat.
Also what does common salt mean?
As it refers to only NaCl, was a concern that if the word ‘salt’ was mentioned maybe it could lead to a misunderstanding, since usually, the salt used for cured products have nitrites/nitrates, which is not the case of this study.
Materials and methods: For normal dry cured ham, heating conditions do not apply. Therefore, the author needs a prior study on how this was manufactured. You also need to accurately describe this for how many batches and repetitions.
Information about batches and repetitions were added to the manuscript.
“Two replications were manufactured at different times and for each replicated lot, three samples of each treatment were randomly selected in a total of 18 bellies and each sample was analyzed in triplicate for each physicochemical analysis.”
Results and discussion:
Line 197-199: It is important to explain the cause of this. We already know about this fact or expectation, but I wonder how to explain this principle in this manuscript.
The responsible characteristic was mentioned.
“(due to the loss of fluids resulting from the hygroscopic characteristic of NaCl)”
The author needs to look at the considerations in the manuscript as a whole. This is very important. I have a hard time seeing most of the reviews as supporting content. It seems that only the analysis of the prediction was made, not the exact principle analysis. Also, when describing conflicting considerations, describe how they differ.
We are sorry for that we are not able to know well how to attend to this requirement. We have revised the whole document and as far as we could, we have assured that conflicting considerations are explained.
Other minor issues include:
Fig. 3. In BG, you can infer from the picture that 13.26 above means T1 and 17.48 means T2, but this picture has a problem. Are there any precedents for this being used?
All of the graphs were modified for simple bar graphs.
Expressing significant differences is insufficient (within figures). Overall, the question arises as to why the figure was produced in this form and not as a simple bar graph.
In the only case that there was a significant difference (water activity) the letters (a,b) were added to identify it correctly.
Expressions such as (BG at 2%) expressed on line 219 need to be rewritten.
It was decided to remove the ‘BG at’ as it can cause confusion among readers, and it was also a repetition of the treatment used, therefore just the concentration of the salt mix was mentioned.
Line 224-226: Is the comparison made because it was conducted under the same conditions as this study?
The authors decided to remove the reference, and therefore this phrase.
Line 235: Do consumers often buy dry-cured ham as whole meat?
Indeed, for dry-cured ham it is harder for consumers to buy the whole piece considering the price and amount of product, although for dry-cured pork bellies it can be exposed in large or small pieces and slices, and the aim of this sentence was to acknowledge the cases where the product is sold in larger pieces.
